# Quality of Life and Psychological Impact in Patients with Atopic Dermatitis

**DOI:** 10.3390/jcm10061298

**Published:** 2021-03-21

**Authors:** Marina Talamonti, Marco Galluzzo, Dionisio Silvaggio, Paolo Lombardo, Chiara Tartaglia, Luca Bianchi

**Affiliations:** 1Department of Systems Medicine, University of Rome “Tor Vergata”, 00133 Rome, Italy; talamonti.marina@gmail.com (M.T.); dionisio.silvaggio@gmail.com (D.S.); lombardo.paolo89@gmail.com (P.L.); chiara-tartaglia@hotmail.it (C.T.); 2Department of “Experimental Medicine”, University of Rome “Tor Vergata”, 00133 Rome, Italy; marco.galluzzo83@gmail.com

**Keywords:** atopic dermatitis, Eczema Area and Severity Index (EASI), alexithymia, depression, quality of life, Beck Depression Inventory (BDI), 20-item Toronto Alexithymia Scale (TAS-20), Dermatology Life Quality Index (DLQI)

## Abstract

Atopic dermatitis (AD) is a dermatological disorder that affects patients’ mental health and psychological state in complex ways. The importance of understanding the entire scope of this burden is well recognized, but there is limited comprehensive information about the resulting stress on adult patients with AD. This study aimed to determine the degree of psychological stress in patients with AD compared to healthy participants. A total of 352 adult patients participated in this cross-sectional study—174 with AD and 178 healthy participants. Demographic and clinical data were collected. Itch and sleep disturbance were assessed using a numeric rating scale and a visual analogue scale. The 20-item Toronto Alexithymia Scale (TAS-20) and Beck Depression Inventory (BDI) questionnaires were administered to assess the symptoms of alexithymia and depression. Quality of life (QOL) was assessed in AD patients using the Dermatology Quality Index. In our study, we found high TAS-20 and BDI scores among patients with AD. The prevalence of alexithymic personality features was 56.3% in patients with AD versus 21.3% in healthy controls (*p* < 0.001). Based on BDI scoring (BDI-21 > 13), depression was suspected in a significantly higher number of patients with AD than in the control group (56.9% (99/174) vs. 15.7% (28/178); *p* < 0.0001). Eczema Area and Severity Index (EASI) score did not show any significant correlations with psychological parameters. Among clinical parameters, only sleep disturbance was positively correlated with depression (R = 0.307, *p* < 0.005). Our data show that the severity index score as a representative factor of skin involvement has a limited role in predicting the effect of skin diseases on mental status. Screening and assessment for psychiatric disorders, QOL, and sleep disturbance in patients with atopic dermatitis cannot be neglected by physicians and they should be treated in clinical practice with the consideration of psychosomatic approaches.

## 1. Introduction

The skin is the largest and most visible part of the human body. Therefore, patients who suffer from undesirable appearance and subjective symptoms due to various skin conditions commonly also experience psychological stress and reduced quality of life (QOL). Such skin diseases include atopic dermatitis (AD), psoriasis, urticaria, alopecia areata, and vitiligo. The psychological aspects of skin disease have been widely investigated, resulting in an increasing emphasis on understanding this facet of skin disease and implementing related treatment approaches [1,2,3,4].

Atopic dermatitis, which is characterized by intense pruritus and very sensitive dry skin, is one of the most common chronic and relapsing inflammatory diseases [5]. Psychiatric comorbidities, including depression, anxiety, and suicidal ideation, are more common in individuals with AD than in the general population, even among patients with clinically mild or moderate disease. In fact, due to their impact on relationships and daily activities, AD and other types of eczema dramatically affect patients’ QOL as assessed through psychological, social, physical, and functional factors [5,6]. 

In particular, mental processes in patients with AD may be relevant in the course of the disease because they could trigger attacks, impact quality of life, and interfere with coping. Despite this, the subject has been little investigated.

## 2. Materials and Methods

### 2.1. Patients and Study Design

A descriptive, cross-sectional, observational study was conducted with patients diagnosed with AD. Patients were recruited from the Department of Dermatology, University of Rome Tor Vergata, in the period from February 2018 to February 2020.

Of the 300 AD patients initially selected, 126 were excluded for the following reasons: 40 did not wish to participate, 56 did not meet the criteria for being diagnosed with AD, and 30 did not answer the questionnaire completely. The inclusion criteria were as follows: a diagnosis of AD lasting one year according to the consensus criteria established by the American Academy of Dermatology [7], patients naïve to any systemic treatment, and age ≥18 years. Exclusion criteria were as follows: other chronic skin disease, serious cognitive impairment, psychiatric disorders, and alcohol and/or drug abuse.

All 174 participants with AD followed the routine protocol used in our AD clinic. As a control group, 178 individuals were randomly selected among healthy persons undergoing routine screening for skin cancer during the same time period, matched for age and gender with the AD patients. Demographic and anamnestic data, comorbidities, and disease severity at the time of enrolment were recorded using a dedicated database. All patients gave written informed consent for their participation prior to enrolment. This studycomplied with the ethical standards laid down in the 1975 Declaration of Helsinki.

### 2.2. Clinician-Reported Outcomes of AD Patients

The disease severity of the AD patients was assessed using the Eczema Area and Severity Index (EASI) [8]. The EASI is a composite score based on the total area affected and intensity of redness, thickness, scratching, and lichenification on the head/neck, trunk, upper limbs, and lower limbs. The score ranges from 0 to 72, with higher scores indicating greater severity.

All patients included in the study were asked to assess their pruritus according to the Pruritus Numerical Rating Scale (NRS-Pruritus; range 0–10) [9,10]. This scale is interpreted as follows: no pruritus (NRS = 0), mild pruritus (NRS 0–2), moderate pruritus (NRS 3–7), severe pruritus (NRS 8–9), and very severe pruritus (NRS > 9) [9,10].

Subjects were also asked to rate their sleep disturbance/deprivation using the visual analogue scale (sleep-VAS) with a range of 0–10 (0: no sleepiness; 10: maximum sleepiness) [11].

### 2.3. Psychometric Assessment

Assessments were performed during clinic visits.

The Toronto Alexithymia Scale (TAS-20), a self-report questionnaire containing 20 items, was used to assess alexithymia [12]. Each item is rated on a 5-point Likert scale ranging from 1 (strongly disagree) to 5 (strongly agree). The TAS-20 also has 3 subscales. The first subscale is difficulty identifying feelings, as in “I have difficulty identifying the correct words for my emotions”. The second subscale is difficulty describing feelings, as in “I have difficulty describing my emotions”. Finally, the third subscale is externally oriented thinking, as in “I would rather talk about people’s daily life activities than their emotions”. The total score ranges from 20 to 100. In order to facilitate comparison with other studies, we assessed the presence or absence of alexithymia using internationally accepted cut-off value, as follows: 20–50, non-alexithymic patients; 51–60, borderline alexithymic patients; and 61–100, alexithymic patients. 

The Beck Depression Inventory (BDI) is a self-reported questionnaire containing 21 items used to evaluate the characteristic symptoms and attitudes of depression. The purpose of this assessment tool is to assess the level of depressive symptoms, including lack of motivation, loss of interest, depressive thoughts, and dysphoric mood. The BDI is scored using a 4-point Likert scale from 0 to 3 for each item; the final score ranges from 0 to 63 points. Higher total scores indicate more severe depression according to the following: normal, 0–12 points; mild depression, 13–18 points; moderate depression, 19–29 points; and severe depression, ≥30 points [13].

In addition, AD patients completed the validated Italian version of the Dermatology Life Quality Index (DLQI) to assess their health-related quality of life [14]. The scoring ranges from 0 (no impact of the disease on QOL) to 30 points (extreme impact of the disease on QOL). 

All patients gave written informed consent for their participation prior to enrolment. This study complied with the ethical standards established in the 1975 Declaration of Helsinki.

### 2.4. Statistical Analysis

Data are presented as mean ± standard deviation for continuous variables and a number and percentage for categorical variables. The χ^2^ or Fisher exact test was used to compare differences between categorical variables, whereas differences between continuous variables were determined by Student’s t-test. An exploratory analysis was also performed to determine the effect of gender, age, and disease severity on psychometric assessments in patient groups. Pearson’s correlation coefficient was used to assess relationships between the parameters studied. *p* < 0.05 was considered statistically significant. All analyses were performed using STATA 11.2 software (StataCorp LP Inc., College Station, TX, USA).

## 3. Results

### 3.1. Patient Demographic and Clinical Characteristics

A total of 352 adult patients participated in this cross-sectional study, divided as follows: 174 (85 men and 89 women) with atopic dermatitis (AD patients) and 178 (90 men and 88 women) healthy participants (controls). The demographics and disease characteristics are presented in Table 1.

The mean age (±SD) of AD patients and controls was 38.1 (±14.0) years (range 18–81) and 39.9 (±13.1) years (range 19–80), respectively. The mean duration of AD was 24.9 ± 13.8 years.

The study population showed a mean EASI score of 27.5 (±11.8). Based on the EASI scoring system, AD severity was classified as moderate (EASI 7–21) in 25.3% (44/174) and severe (EASI > 21) in 74.7% (130/174) of AD patients. The mean pruritus intensity according to the NRS was 7.7 points (±2.2), while the mean sleep-VAS was 6.2 (±3.0).

### 3.2. Alexithymia

The mean TAS-20 score was 52.3 (±12.9) for the AD patients, indicating that the group of patients as a whole had a score close to the cut-off point for an alexithymia diagnosis (≥61 points); the corresponding score was 45.1 (±10.8) for the control group (*p* < 0.0001). The alexithymic characteristics of the study participants are presented in Table 2.

The prevalence of alexithymic personality features (TAS-20 ≥ 51) was 56.3% in patients with AD versus 21.3% in healthy controls (*p* < 0.001). In the AD group, 58 (33.3%) patients were classified as alexithymic (TAS-20 score ≥ 61) and 40 (23.3%) were classified as borderline, while a total of 18 (10.1%) controls were classified as alexithymic and 20 (11.2%) subjects were classified as borderline. 

### 3.3. Depression

The mean total BDI-21 score was significantly worse for AD patients than for controls (15.4 ± 10.2 vs. 6.8 ± 4.8, respectively; *p* < 0.0001). Based on BDI scoring (BDI-21 > 13), depression was suspected in a significantly higher number of AD patients than in the control group (56.9% (99/174) vs. 15.7% (28/178); *p* < 0.0001). In particular, 23.0% of AD patients reported moderate depression, and 12.1%. reported severe depression. The details of the BDI-21 results for each group are shown in Table 2. 

### 3.4. AD Patient Reports

The DLQI score for AD patients ranges from 0 to 30 points (mean 13.8 ± 7.0 points; Table 3). Based on the DLQI categorization [14], only two (1.1%) AD patients reported no negative influence of skin disease on their QOL, while 26 (14.9%) reported a small effect (2–5 points), 36 (20.7%) reported a moderate effect (6–10 points), and 110 (63.3%) reported a very large effect (≥11 points) [15].

DLQI scores by sex are also reported in Table 3. The data show that men had a mean score of 13.2 (±7.8) while women had a mean score of 14.4 (±6.2); there is no significant difference, therefore, regarding DLQI scoring between men and women. 

Patient reports on pruritus intensity showed a mean NRS-Pruritus score of 7.7 (±2.2). Itching had the highest score in both men and women, with a total of 61 men (71.8%) and 62 women (69.7%) reporting severe or very severe pruritus (Table 4).

The prevalence of alexithymic features (borderline or full-blown alexithymia) was higher among women (61.8%) than among men (50.6%), but the difference is not statistically significant (*p* = 0.1690); the prevalence of alexithymia was similar among men (33.0%) and women (33.7%) with AD, *p* = 1.0000 (Table 5).

With regard to mild-to-moderate AD versus moderate-to-severe AD (EASI score < 16 or ≥16), no significant differences were found regarding any of the psychometric parameters (TAS-20, BDI) considered—sleep-VAS, NRS-Pruritus, and QOL scores (Table 6). 

Furthermore, there were no statistically significant differences in mean score among patients stratified by age (<40 or ≥40 years) and disease duration (<15 and ≥15 years) for any of the parameters considered (Table 6).

### 3.5. Pearson’s Correlation Analysis

The relationships between age, clinical parameters (i.e., disease duration, EASI, NRS-Pruritus, sleep-VAS), and QOL, TAS-20, and BDI are presented in Table 7. 

The results show that NRS-Pruritus is strongly correlated with sleep disturbance (r = 0.5621, *p* < 0.00001) and weakly to moderately correlated with disease duration (r = 0.2665, *p* < 0.01) and EASI score (r = 0.3656, *p* = 0.000093). 

Furthermore, the data also show that sleep disturbance is correlated with DLQI score (r = 0.2992, *p* < 0.005) and BDI score (r = 0.307, *p* < 0.005). 

We also noted a strong correlation between TAS-20 score and BDI score (*p* < 0.00001). 

Patient age and other clinical parameters (EASI and disease duration) were not shown to be significantly correlated with DLQI or psychometric scores such as TAS-20 and BDI. 

## 4. Discussion

Atopic dermatitis often causes constant, intense itching, highly visible symptoms (e.g., redness, flaking, and bleeding due to scratching), and reduced psychosocial interaction and quality of life [16,17]. It is well known that AD exerts a detrimental effect on the lives of patients and their families, including social, academic, occupational, and financial burdens, overall severely impacting their quality of life [18]. 

Dramatic advances in the treatment of atopic dermatitis have increased the need for severity and QOL evaluation in clinical research and practice. There are many available measures to assess the severity of AD, but key symptoms such as pruritus, sleep disturbance, and interference with activities are difficult to assess. Furthermore, physicians often tend to underestimate the intensity of symptoms [16].

This study, which reflects real-world clinical settings, focused on the impact of AD symptoms on adult patients’ daily lives and psychological well-being. 

In a large population (174 patients vs. 178 controls), we found high TAS-20 and BDI scores among patients with AD. In line with previous research, 33.3% of the patients suffer from alexithymia and can be considered as a psychologically at-risk group [19].

In this study, BDI score, which indicates the state of depression, was slightly higher in AD patients than in healthy individuals. Based on BDI scoring (BDI-21 > 13), the prevalence of depression (56.9% in the AD group vs. 15.7% in the controls) falls within the range of worldwide studies [20,21].

A recent Italian study reported that alexithymia was more common among patients with severe AD than patients with mild AD and that it correlated with itch intensity and sleep disturbances. However, as reported by the authors, their study included both untreated patients and patients undergoing therapy [21]. On the contrary, our study included only AD patients naïve to systemic treatments, and the EASI score did not show any significant correlations with psychological parameters. Furthermore, patient age and other clinical parameters (EASI and disease duration) were not shown to be significantly correlated with DLQI or psychometric scores such as TAS-20 and BDI. 

Among clinical parameters, only sleep disturbance was positively correlated with depression (R = 0.307, *p* < 0.005). 

Previous studies have shown an increased prevalence of depression in both children and adults when compared with their healthy peers. In children, a positive association was found between AD and depression, although the estimate was weaker than that for adults [22]. 

Our data fall in line with studies that have shown that in chronic inflammatory skin disorders, including psoriasis and hidradenitis suppurativa, the severity index score as a representative factor of skin involvement has a limited role in predicting the effect of the disease on the patient’s mental state [23,24,25,26].

On the other hand, we found a correlation between subjective symptoms (pruritus and sleep disturbance) and clinical parameters, i.e., the extent and intensity of AD and disease duration. 

NRS-Pruritus was strongly correlated with sleep disturbance (r = 0.5621, *p* ≤ 0.00001) and weakly to moderately correlated with disease duration (r = 0.2665, *p* = 0.009561) and EASI score (r = 0.3656, *p* = 0.000093). 

As is commonly believed, many authors have reported that the intense itching associated with AD often causes patients to experience severe sleep disturbance, leading to daytime sleepiness and sleep-related impairment [27,28,29]. Sleep disturbance consequently results in functional impairment and profoundly worsens QOL for AD patients [30,31]. It is also associated with unsatisfactory performance in school and work, reduced general health and safety, and considerable cost [32]. 

Related to this, a large body of evidence has demonstrated the vital role that sleep plays in maintaining health [33,34,35,36,37] and optimal physiological [35] and psychological performance [38]. There is also evidence that sleep disruption may increase the risk of psychological illness (e.g., depression and anxiety) [39].

Finally, when investigating QOL, we noted an association between sleep disturbance (*p* < 0.005), TAS-20 score (*p* < 0.05), and BDI score (*p* < 0.005). Our data confirm that QOL is associated with sleep disturbance and a high risk of depression, so patients with poor QOL and/or sleep disturbance represent a subgroup of patients with specials needs.

## 5. Conclusions

AD has a significant impact on psychosocial well-being and quality of life, including the presence of daily pruritus-associated effects such as sleep disturbance and functional impairment and secondary consequences including psychiatric issues. In line with previous studies in adult patients with AD, our data reveal no correlation between disease severity and QOL and psychiatric disorders, suggesting that even mild-to-moderate forms of AD can be enough to adversely affect emotional condition and personality [38]. The clear relationships found between sleep disturbance, QOL, and BDI and a significant correlation between QOL, TAS-20, and BDI can be considered important knowledge gained from this study. Itching is likely responsible for sleep disturbances, thereby causing psychological distress, which may result in the development of depression and/or impaired QOL. QOL assessments that include data regarding psychological impact are, therefore, increasingly showing great potential in capturing the more holistic impact of itching. 

Further research is needed to characterize and define subpopulations at increased risk of psychological stress (e.g., adolescents, children younger than 12 years, and adult-onset AD).

Furthermore, screening and assessment for QOL, sleep disturbance, and depression in patients with atopic dermatitis cannot be neglected by physicians and AD should be treated in clinical practice in consideration of psychosomatic approaches. It remains to be investigated whether the alexithymic features and depressive symptoms result from physical discomfort, i.e., from itching, sleeplessness, other stress factors, and the psychosocial burden of the skin disease, or whether these disorders are based on shared inflammatory pathomechanisms.

## Figures and Tables

**Table 1 jcm-10-01298-t001:** Demographics and disease characteristics.

	AD Patients	Healthy Population
Number	174	178
Males/females	85/89	90/88
Age, mean ± SD (range)	38.1 ± 14.0 (18–81)	39.9 ± 13.1 (19–80)
Duration of AD, mean ± SD (range)	24.9 ± 13.8 (3–75)	–
EASI, mean ± SD (range)	27.5 ± 11.8 (3–55)	–
NRS-itch, mean ± SD (range)	7.7 ± 2.2 (0–10)	–
Sleep-VAS, mean ± SD (range)	6.2 ± 3.0 (0–10)	–
Comorbidities, N (%)	27 (15.5%)	31 (17.4%)

AD: Atopic Dermatitis; EASI: Eczema Area and Severity Index; NRS: Numeric Rating Scale.

**Table 2 jcm-10-01298-t002:** Toronto Alexithymia Scale (TAS-20) and Beck Depression Inventory (BDI) continuous and categorical data.

	AD Patients	Healthy Population	*p* *
**TAS-20**, mean ± SD (range)	52.3 ± 12.9 (20–75)	45.1 ± 10.8 (20–74)	<0.0001
Non-alexithymic, N (%)	76 (43.7%)	140 (78.7%)	<0.0001
Borderline alexithymic, N (%)	40 (23.0%)	20 (11.2%)	<0.0001
Alexithymic, N (%)	58 (33.3%)	18 (10.1%)	<0.0001
**BDI**, mean ± SD (range)	15.4 ± 10.2 (0–39)	6.8 ± 4.8 (0–20)	<0.0001
Normal, N (%)	75 (43.1%)	150 (84.3%)	<0.0001
Depression symptoms	99 (56.9%)	28 (15.7%)	<0.0001
Mild depression, N (%)	38 (21.8%)	22 (12.4%)	0.0229
Moderate depression, N (%)	40 (23.0%)	3 (1.7%)	<0.0001
Severe depression, N (%)	21 (12.1%)	3 (1.7%)	<0.0001

* *p*-Values < 0.05 were considered statistically significant.

**Table 3 jcm-10-01298-t003:** Dermatology Life Quality Index (DLQI) score—continuous and categorical data of AD.

	AD Group	Men	Women	*p* *
**DLQI**, mean SD (range)	13.8 ± 7.0 (1–30)	13.2 ± 7.8 (1–30)	14.4 ± 6.2(3–30)	0.2644
DLQI score < 2	2 (1.1%)	2 (2.4%)	0 (0.0%)	0.2456
DLQI 2–5 score, N (%)	26 (14.9%)	15 (17.6%)	11 (12.4%)	0.3969
DLQI 6–10 score, N (%)	36 (20.7%)	19 (22.4%)	17 (19.1%)	0.7086
DLQI >10 score, N (%)	110 (63.3%)	49 (57.6%)	61 (68.5%)	0.1582

* *p*-Values < 0.05 were considered statistically significant.

**Table 4 jcm-10-01298-t004:** Pruritus Numerical Rating Scale (NRS) score—continuous and categorical data of AD.

	AD Group	Men	Women	*p* *
**NRS-Pruritus**	7.7 ± 2.2 (0–10)	7.5 ± 2.5 (0–10)	7.9 ± 2.0 (2–10)	0.3331
Mild pruritus (NRS 1–2)	19 (10.9%)	8 (9.4%)	11 (12.4%)	0.6298
Moderate pruritus (NRS ≥ 3 < 7)	32 (18.4%)	16 (18.8%)	16 (18.0%)	1.0000
Severe pruritus (NRS ≥ 7 < 9 )	57 (32.8%)	25 (29.4%)	32 (36.0%)	0.4199
Very severe pruritus (NRS ≥ 9)	66 (37.9%)	36 (42.4%)	30 (33.7%)	0.2753

* *p*-Values < 0.05 were considered statistically significant.

**Table 5 jcm-10-01298-t005:** TAS–20 and BDI scores in men and women of AD group.

	Men	Women	*p* *
**TAS-20** mean ± SD, (range)	52.0 ± 12.1 (20–74)	52.7 ± 12.2 (24–75)	0.7259
Non-alexithymic, N (%)	42 (49.4%)	34 (38.2%)	0.1247
Borderline alexithymic, N (%)	15 (17.6%)	25 (28.1%)	0.1089
Alexithymic, N (%)	28 (33.0%)	30 (33.7%)	1.0000
**BDI** mean ± SD, (range)	14.9 ± 9.7 (0–38)	15.8 ± 10.7 (0–39)	0.6004
Normal, N (%)	33 (43.1%)	42 (47.2%)	0.2867
Mild depression, N (%)	23 (21.8%)	15 (16.9%)	0.1415
Moderate depression, N (%)	21 (23.0%)	19 (21.3%)	0.7189
Severe depression, N (%)	8 (12.1%)	13 (14.6%)	0.3553

* *p*-Values < 0.05 were considered statistically significant.

**Table 6 jcm-10-01298-t006:** Relationship between psychometric assessments and selected clinical parameters.

	N (Males/Females)	NRS-SLEEP	NRS-ITCH	DLQI	TAS-20	BDI
**EASI < 16**	50	5.5 ± 2.2	7.5 ± 1.9	12.3 ± 5.5	52.9 ± 11.9	14.5 ± 10.4
	(21/29)	(0–10)	(2–10)	(3–24)	(24–74)	(0–36)
**EASI ≥ 16**	124	7.9 ± 2.2	6.4 ± 2.9	14.4 ± 7.4	52.9 ± 11.9	16.0 ± 10.0
	(65/59)	(1–10)	(0–10)	(0–30)	(24–74)	(0–39)
*p*	-	0.0622	0.4052	0.0879	0.5484	0.3669
**Age ≥ 40**	71	5.7 ± 3.2	7.5 ± 2.6	13.8 ± 6.7	53.9 ± 13.3	16.0 ± 11.1
	(25/46)	(0–10)	(1–10)	(0–30)	(26–75)	(0–39)
**Age < 40**	103	6.59 ± 2.8	7.8 ± 1.9	13.8 ± 7.3	51.2 ± 12.5	14.9 ± 9.5
	(60/43)	(1–10)	(0–10)	(1–30)	(20–74)	(0–36)
*p*	-	0.1937	0.2665	0.9507	0.1741	0.4806
**AD duration < 15**	70	6.7 ± 2.6	7.8± 2.4	14.9± 6.8	58.3 ± 11.3	17.6± 10.0
	(22/48)	(1–10)	(1–10)	(2–30)	(30–73)	(3–38)
**AD duration ≥ 15**	104	5.9 ± 3.0	7.4 ± 2.3	14.5 ± 7.3	51.2 ± 12.5	15.1 ± 10.2
	(50/64)	(0–10)	(0–10)	(2–30)	(20–74)	(0–36)
*p*		0.3108	0.5287	0.8270	0.1741	0.3701

**Table 7 jcm-10-01298-t007:** Correlations between psychometric assessments and the clinical parameters studied.

	Age *R*(*p*)	AD Duration *R*(*p*)	EASI *R*(*p*)	NRS-Pruritus *R*(*p*)	VAS-Sleep *R*(*p*)	DLQI *R*(*p*)	TAS-20 *R*(*p*)	BDI *R*(*p*)
**Age**	-	0.1176	0.0541	0.0166	0.0336	0.0704	0.042	0.1414
		(0.269386)	(0.585447)	(0.864604)	(0.735787)	(0.507258)	(0.68452)	(0.167117)
**Disease duration**	0.1176	-	**0.278**	**0.2665**	**0.2711**	0.0846	0.1658	0.1512
	(0.269386)		(0.0087298)	**(0.009561)**	**(0.00937)**	(0.410002)	(0.113984)	(0.184061)
**EASI**	0.0541	**0.278**	-	**0.3656**	**0.2749**	0.1993	0.0126	0.2073
	(0.585447)	**(0.0087298)**		**(0.000093)**	**(0.00937)**	(0.062661)	(0.907246)	(0.052629)
**NRS-Pruritus**	0.0166	**0.2665**	**0.3656**	-	**0.5621**	0.0944	0.0358	0.0997
	(0.864604)	(0.009561)	**(0.000093)**		**(<0.00001)**	(0.362848)	(0.73193)	(0.336407)
**VAS-sleep**	0.0336	**0.2711**	**0.2749**	**0.5621**	-	**0.2992**	0.1785	**0.307**
	(0.735787)	(0.00937)	(0.00937)	**(<0.00001)**		(0.003396)	(0.085193)	**(0.002615)**
**DLQI**	0.0704	0.0846	0.1993	0.0944	**0.2992**	-	**0.1874**	**0.306**
	(0.507258)	(0.410002)	(0.062661)	(0.362848)	**(0.003396)**		(0.040843)	**(0.001279)**
**TAS-20**	0.042	0.1658	0.0126	0.0358	0.1785	**0.1874**	-	**0.5845**
	(0.68452)	(0.113984)	(0.907246)	(0.73193)	(0.085193)	**(0.040843)**		**(<0.00001)**
**BDI**	0.1414	0.1512	0.2073	0.0997	**0.307**	**0.306**	**0.5845**	-
	(0.167117)	(0.184061)	(0.052629)	(0.336407)	(0.002615)	(0.001279)	**(<0.00001)**	

Bold number define *p*-values < 0.05.

## Data Availability

Data is contained within the article.

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
