# Peer review of "Quality of Life and Psychological Impact in Patients with Atopic Dermatitis"

_jcm, 2021, doi:10.3390/jcm10061298_

Round 1

Reviewer 1 Report

It is interesting and well written work that clarify the importance of screening and assessment for QOL, sleep disturbance, and depression in patients with atopic dermatitis. The authors showed that itching is likely responsible for sleep disturbances, thereby causing psychological distress, which may result in the development of depression and/or impaired QOL. So they conclude that atopic dermatitis should be treated in clinical practice in consideration of psychosomatic approaches. 

These results add important data to the world literature, so I believe the paper can be accepted as it is.

Author Response

Dear reviewer,

thank you for your time and opportunity in revising our work.

We greatly appreciate your comment.

Reviewer 2 Report

Different similar studies are already present in the literature. Given the very high number of participants and comparison, I think the authors should in my opinion do an ANOVA test instead. No correction for multiple comparison was performed, such as a Bonferroni test.

I think the authors should modify the statistics and then resubmit the paper.

Article is not at the moment publishable.

Author Response

Response to Reviewer 2 Comments

1)      Different similar studies are already present in the literature.

Response: We thank this reviewer for their comment.

As reported in the Discussion and Conclusion section, QoL impairment associated with AD is demonstrated by the spectrum of psychiatric disorders that may be observed in AD  patients, such as depression and anxiety (Eckert L et al.. Impact of atopic dermatitis on health-related quality of life and productivity in adults in the United States: An analysis using the National Health and Wellness Survey. J Am Acad Dermatol. 2017; 77(2):247-279; Cheng BT Silverberg JI Depression and psychological distress in US adults with atopic dermatitis Ann Allergy Asthma Immunol. 2019; 123: 179–185

Few studies in which an association between AD and alexithymia has been reported; our study aimed to value association between Alexythimia, depression symtomps and QoL expanding the spectrum of psychological disorders associated with AD

2)      Given the very high number of participants and comparison, I think the authors should in my opinion do an ANOVA test instead. No correction for multiple comparison was performed, such as a Bonferroni test.

Response: We thank the reviewer for their suggestions.

As reported , we performed the χ2 or Fisher exact test to compare differences between categorical variables, whereas differences between continuous variables were determined by Student’s t-test.

The Student's t test was used to compare exactly our two groups  ( AD group versus control group), whereas a  statistical technique used to compare the means between three or more groups is known as ANOVA or F test.  (Mishra, Prabhaker et al. “Application of student's t-test, analysis of variance, and covariance.” Annals of cardiac anaesthesia vol. 22,4 (2019): 407-411. doi:10.4103/aca.ACA_94_19).

In the next study, we will evaluate the same tests in many groups of patients treated with different therapies, and we will use the suggested tests.

Reviewer 3 Report

I would suggest including a comment in the discussion paragraph regarding the disease duration (the mean duration of AD was almost 25 years in this study). We are talking about adult patients that had AD for quite some time. One can ask if depression or alexithymia are significant issues even for pediatric patients. Sometimes discussions and conclusions are written to generally. It is not clear specified that we are talking only about adults with long disease history. Based on results in this study we cannot generalyze to all AD patient groups.

Round 2

Reviewer 2 Report

The authors responded to all queries.